# Sequence Segmentation of Nematodes in Atlantic Cod with Multispectral Imaging Data

**DOI:** 10.3390/foods13182952

**Published:** 2024-09-18

**Authors:** Andrea Rakel Sigurðardóttir, Hildur Inga Sveinsdóttir, Nette Schultz, Hafsteinn Einarsson, María Gudjónsdóttir

**Affiliations:** 1Faculty of Food Science and Nutrition, University of Iceland, Sæmundargata 12, 102 Reykjavík, Iceland; ars59@hi.is (A.R.S.); hilduringa@matis.is (H.I.S.); 2Matís, Food and Biotech R&D, Vínlandsleið 12, 113 Reykjavík, Iceland; 3Videometer A/S, 2730 Herlev, Denmark; 4Faculty of Computer Science, University of Iceland, Bjargargata 1, 102 Reykjavík, Iceland; hafsteinne@hi.is

**Keywords:** multispectral imaging, fish fillet inspection, image processing, nematode detection

## Abstract

Nematodes pose significant challenges for the fish processing industry, particularly in white fish. Despite technological advances, the industry still depends on manual labor for the detection and extraction of nematodes. This study addresses the initial steps of automatic nematode detection and differentiation from other common defects in fish fillets, such as skin remnants and blood spots. VideometerLab 4, an advanced Multispectral Imaging (MSI) System, was used to acquire 270 images of 50 Atlantic cod fillets under controlled conditions. In total, 173 nematodes were labeled using the Segment Anything Model (SAM), which is trained to automatically segment objects of interest from only few representative pixels. With the acquired dataset, we study the potential of identifying nematodes through their spectral signature. We incorporated normalized Canonical Discriminant Analysis (nCDA) to develop segmentation models trained to distinguish between different components within the fish fillets. By incorporating multiple segmentation models, we aimed to achieve a satisfactory balance between false negatives and false positives. This resulted in 88% precision and 79% recall for our annotated test data. This approach could improve process control by accurately identifying fillets with nematodes. Using MSI minimizes unnecessary inspection of fillets in good condition and concurrently boosts product safety and quality.

## 1. Introduction

Process automation is a significant part of technical advancements in the modern fishing industry [1,2]. Companies can improve efficiency, reliability, and precision by automating tasks with high-tech solutions developed for fish processing [3]. Today, there are cutting-edge automated solutions for many steps during fish processing, ranging from grading [4] and de-heading [5], to intricate steps such as inspection and removal of pin bones [6], portioning [1,7], and packaging [8].

Atlantic cod (*Gadus morhua*) is among the most important commercial fish species on the world market, and it is one of the most popular species among EU consumers [9]. For sellers, the quality of cod fillets is of utmost importance as it directly impacts customer satisfaction, repeat business, and the overall reputation of the product. Nematodes, a type of parasitic worm in fish, pose a substantial challenge for the fish processing industry [10]. Traditional detection and removal of nematodes relies on manual labor, but advancements in computer vision and deep learning have enabled further research into more efficient methods. The anisakis nematodes most commonly found in Atlantic cod are *Anisakis simplex sensu stricto* and *Pseudoterranova decipiens*. The infection levels and species diversity of nematodes in Atlantic cod have been shown to correlate with both geographic area and fish size [11]. Nematodes can be different in size and color, *A. simplex* is smaller and has clear coloration, while *P. decipiens* tends to be larger and darker. While consuming a live parasitic nematode can lead to infections, these are typically not a concern for public health if the fish undergo proper cooking or freezing processes that render the parasites noninfectious [12]. Removing nematodes is also economically important, as the presence of nematodes can impact the market value of fish products due to consumer aversion [13]. This underscores the importance of nematode removal as a quality assurance issue in fresh fish.

Visual inspection is the only non-destructive method for nematode detection available for the fish processing industry [12]. After filleting and skinning, cod fillets are inspected on a candling table, a back-lit station where workers manually remove any visible nematodes with a knife [10]. More recent research has shown that the efficiency of candling is about 51% for *Anisakis* detection, as candling was not able to detect all *Anisakis* spp. larvae present in the fillets [11]. Another recent study showed that candling only provided an overall sensitivity of detecting *Ascaridoids*, such as *Anisakis*, as low as 31% [10]. The same study thus called for the development of more in-depth assessments and nematode detection methods.

Since the last EFSA report published in 2010 [13], artificial digestion and the UV-press method have been developed into the ISO standards to detect parasites in fish [14,15] and have been used in large scientific surveys [12,16]. The performance of both methods has shown high accuracy, sensitivity, and specificity for detecting nematodes [17]. However, both methods are destructive to the tissues and time-consuming, and would therefore not be suitable for use during fresh fish processing, as studied in this case [17,18].

Various techniques have been proposed and evaluated for non-destructive automated detection of nematodes in fish, with some presenting encouraging results in pieces of fish in small-scale laboratory settings [19]. A promising technique that has been explored is the detection of nematodes through images acquired by Hyperspectral Imaging (HSI)/Multispectral Imaging (MSI) systems. These techniques have been shown to effectively detect nematodes embedded up to about 8 mm into the Atlantic cod muscle using a combination of spectral bands in the Near-Infrared (NIR) and the visible light range [19]. Additionally, a study using a Hyperspectral Imaging (HSI) system operating at a conveyor belt speed of 40 cm/s, achieved a detection rate of 70.8% of dark nematodes and 60.3% of pale nematodes [20]. However, there were one or more false positive predictions in 60% of the fillets, highlighting how difficult it can be to distinguish the nematodes from the Atlantic cod muscle tissue.

Developing automating nematode detection further presents an opportunity to improve online process control in the fish processing industry, potentially leading to increased product safety, as well as improved production rates and efficiency. By accurately identifying fillets with nematodes and their location within the fillets, operators could streamline the processing workflow. This leads to increased product safety and quality, reduced reliance on manual labor, and improved overall productivity during processing. Additionally, automation can prevent unnecessary examination of fillets that are in good condition and would otherwise need considerable inspection time. Time can be saved by examining a sample of fillets from a batch of suspected fish to assess the level of infestation. This assessment can then determine if the entire batch requires candling or if it is better suited for other purposes [13]. By using an automated method, this process could also become more efficient and accurate, ensuring that only fillets that require further inspection are automatically flagged, thereby optimizing the workflow and reducing manual effort.

This study uses MSI technology to automatically detect nematodes and distinguish them from other common defects found in fish fillets. Our methodology integrates normalized Canonical Discriminant Analysis (nCDA) into the development of a segmentation model, trained explicitly for differentiating between different components within the fish fillets. The nCDA is a supervised model based on an MSI transformation of the images. By incorporating multiple segmentation models and exploiting their collective strengths, individual models may capture the specific characteristics within the fish fillet being segmented.

## 2. Materials and Methods

### 2.1. Multispectral Imaging Equipment

The Multispectral Imaging (MSI) system, VideometerLab 4 (Videometer A/S, Herlev, Denmark), was used to capture multispectral images of the cod fillets. This system combines a high-resolution charged-coupled device (CCD) camera and computer technology with advanced digital image analysis. The instrument uses a strobe light-emitting diode (LED) technology to combine 19 different wavelengths, ranging from 365 to 970 nm, into a single high-resolution spectral image, where every pixel is represented as a reflectance spectrum. The system was calibrated according to color, geometry, and self-illumination, and was set up to operate in 100% reflection mode. The size of the obtained MSI images was 2992 × 2992 pixels. The VideometerLab 4 covers the visible light range and takes advantage of spectral regions outside the visible spectral range, including parts of the ultraviolet (UV) and near-infrared (NIR) regions. This allows extracting additional information that the human eye cannot detect. The signal from an MSI for a given substance is often called a spectral signature [21]. The unique spectral signature of substances allows for easier identification and separation of substances that are visually indistinguishable to the human eye. Additionally, the VideometerLab Software (Videometer A/S, Herlev, Denmark) was used in this study for the Principal Component Analysis (PCA, Section 2.3), modeling (Section 2.4), morphological filtering (Section 2.5), and image segmentation (Section 2.6).

### 2.2. Sample Preparation and Data Acquisition

Atlantic cod (*Gadus morhua*) fillets, 50 in total, were handpicked from the production process at Vísir hf in Grindavík, Iceland, on the 4 October 2021, over a 15-min period. Out of the 50 fillets, 45 were randomly handpicked before the fillets reached the candling tables for trimming, and upon selection, the fillets were inspected to verify the presence of nematodes at various locations. Then, 5 fillets were picked following the trimming station, where they had undergone inspection, and all visible nematodes were manually removed. These fillets were maintained as a quality assurance control group.

To ensure a consistent and controlled lighting environment, the fillets were cut to fit the narrow diameter (110 mm) of the VideometerLab dome. The cuts included loins, belly flaps, middle sections, and tailpieces, resulting in 270 individual fish pieces. No further preparations were made to the fillets prior to imaging.

### 2.3. Manual Nematode Annotation

Principal Component Analysis (PCA) was performed on all images as an image enhancement to assist with image annotations and data visualization. The main advantage of using PCA for data visualization is that it can handle high-dimensional data sets that are difficult to plot or interpret otherwise. PCA can reduce the data’s complexity and noise, highlighting the most important features and relationships [22]. For image annotation, the third principal component (PC 3) was specifically chosen because nematodes were visually clearer in this component compared to others. The enhanced visibility of the nematodes in PC 3 made it a suitable blueprint for labeling, while other principal components were referenced for additional context during the annotation process. Figure 1 illustrates the difference between a standard RGB image and the PC 3 images, demonstrating how PC 3 highlights the nematodes effectively for annotation purposes.

The Computer Vision Annotation Tool (CVAT, www.cvat.ai, CVAT.ai corporation, Pathos, Cyprus) was used to create the annotations for the dataset. The Segment Anything Model (SAM) [23] was used for assistance in the labeling process. When using SAM, an object can be segmented with only a few clicks on representative pixels for the object of interest. SAM was trained to have a general understanding of what objects are likely to be present. Originally, the SAM was trained by Meta AI (www.meta.ai (accessed on 30 May 2024)) to have a general understanding of what objects are likely to be present in an image. An example of the process is presented in Figure 2. This capability allows the SAM to recognize unfamiliar objects without needing any further training, and because of its zero-shot generalization ability, the SAM was selected for labeling. Nematodes were the only class objects labeled, and 173 were labeled. Labels were exported as ground truth masks.

The distribution of nematodes in the different parts of the fish fillets is shown in Table 1. The table shows the average number of nematodes present in images that belong to different parts of the fish fillet. The belly flap had the highest average number of nematodes, which was expected as this area is the most likely to be affected [10,11].

### 2.4. nCDA

Normalized Canonical Discriminant Analysis (nCDA) is a supervised model based on MSI transformation of images that determines how to best separate or discriminate two or more groups of individuals, given quantitative measurements of several variables related to these individuals. Given a desired categorical variable and several independent variables that are unwanted, they should be distinguishable from the desired categorical variable. The nCDA determines the relationship between them by deriving canonical variables, linear combinations of the unwanted variables that encapsulate the variation between the selected classes, similar to a PCA, which summarizes the total variation. nCDA thus maximizes variation between individual groups while minimizing the original variables’ within-group variation [24].

In this study, nCDA was used to separate different fish fillet components that may seem similar to our eyes, such as distinguishing fish muscle from clear or yellow nematodes, as well as veins and blood from nematodes that are reddish in color. Two different kinds of nCDA transformations were made, using 54 images as training data, and 216 images for test data. Each image was 2992 × 2992 pixels, but only a subset of representative pixels for each class (e.g., nematodes, fish muscle, blood, bruises, and skin) was manually selected for training. Since the number of nematodes and the pixels representing them was limited, this separation was chosen to ensure more representative results. The remaining pixels in the training images, which were not used for training, were used for validation to confirm the model’s performance before applying it to the test set. 

In transformation A, the nematodes were distinguished from the fish muscle, blood, and bruises. In transformation, B, the skin was distinguished from every other element in the fish fillet. This includes nematodes, blood, bruises, and the fish muscle (Figure 3).

### 2.5. Morphological Filtering

For the two segmentation models made, A and B, the morphological filters dilation, erosion, and opening were used with a filter of size 5 × 5 pixels. Morphological filters modify the boundaries of objects [25]. Dilation expands the boundaries of objects by adding pixels, making objects more pronounced, and filling in small gaps. This results in thicker lines and enlarged shapes. Conversely, erosion reduces the boundaries by removing pixels, eliminating extraneous pixels and fine lines to leave only substantial objects. The outcome is thinner lines and smaller shapes. The extent of pixel addition or removal depends on the size and shape of the structuring element used. The opening operation erodes an image and then dilates the eroded image, using the same structuring element for both operations. Opening operations are used to eliminate objects that are small or thin while preserving the shape and size of larger objects in the image.

Morphological filters for images consist of non-linear operations related to the shape or morphology of features in an image [26]. Morphological filtering is filtering transformations that use the values of pixels in a moving window set, W, around the input pixel to compute the output pixel values. An important concept in mathematical morphology includes using sets to represent images and set operations to represent image transformations which were performed as described in [27,28].

### 2.6. Sequence Segmentation

The next step involved a semantic subtraction, where we removed all predicted pixels from model B from the predicted pixels from model A. Figure 4 illustrates the relations between models A and B.

Skin remnants and darker nematodes may appear similar in images, leading to instances where model A incorrectly classifies pixels associated with skin remnants as nematodes due to their resemblance to darker nematodes. The two segmentations, A and B, were sequenced in a particular order by applying bitwise operations like the following:A\B(1)

In cases where model A mistakenly identifies pixels of skin remnants as nematodes, model B is trained to recognize these pixels as skin remnants. Subsequently, these correctly identified skin remnant pixels are excluded from the final segmentation.

### 2.7. Performance Metrics

Semantic segmentation assigns a category label to each pixel of an image. Semantic segmentation tasks, therefore, aim to predict a mask that indicates where the object of interest is present [29]. In this study, there is one type of object to predict, i.e., nematodes. This type of task is binary and the classes included are the background and nematodes. The background class represents every element of the image except for nematodes. The background class consists of the vast majority of the pixels in the image. Therefore, the dataset is highly imbalanced as the nematode pixels represent only a small percentage of the whole image. To manage this imbalance, more images were included in the test set than in the training set. This approach introduced more variability within the test set, providing a better understanding of the model’s performance in detecting different nematodes.

Precision, recall, Intersection over Union (IoU), and *F*1 score are the performance metrics reported to evaluate the performance of the models in this binary pixel classification task. All performance metrics were implemented in Python (version 3.10, Python Software Foundation, Wilmington, DE, USA).

IoU measures the overlap between the predicted segmentation (PM) and the ground truth masks (GT). It is an important metric for evaluating segmentation models because it measures how well the model can separate objects from their background in an image. We can define IoU as the following [30], where PM, GT ⊆ S ∈ R^n^:(2)$$IoU=\frac{\left|PM\cap GT\right|}{\left|PM\cup GT\right|}$$

Although IoU provides an idea of how well the model is performing, it does not fully reflect the localization performance of the model. The weakness of the IoU arises if |PM ∩ GT| = 0 because then IoU(PM, GT) = 0. In this case, the IoU does not reflect whether the two regions, PM and GT, are in proximity to each other or not.

The goal is to determine the number of correctly identified nematode areas using our model. This involves identifying specific objects in the original reference masks (ground truth) and in the model’s predicted masks. The function *Scipy.ndimage.label* [31] is used to identify unique regions by using binary masks as input. Any non-zero values in the input mask are considered features, while zero values are seen as the background. A structuring element was generated that considered features connected even if they touch diagonally:[[1, 1, 1],
[1, 1, 1],(3)
[1, 1, 1]]

The areas in the predicted mask are then compared with those in the ground truth mask. If there is any overlap (an IoU score above a determined threshold) between the two areas, it is considered a correct identification or a true positive. However, there are instances where multiple predicted areas might overlap with the same true area. To prevent counting this as multiple correct predictions, a rule is applied: if a true area has already been paired with a predicted one, any further matches with other predicted areas will not be counted again (and will, in particular, not be counted as something else than a true positive). This ensures that there is no overestimation of the number of correct identifications. By counting the occurrences of true positives (*TP*), false positives (*FP*), and false negatives (*FN*), the precision and recall are defined as follows [30]:(4)$$Precision=\frac{TP}{TP+FP}$$
(5)$$Recall=\frac{TP}{TP+FN}$$

The *F*1 score is defined as the harmonic mean of precision and recall. *F*1 scores range between [0, 1], with 1 representing a model that perfectly classifies each prediction into the correct class. Integration of precision and recall into a single metric provides a better understanding of the model’s performance. A significant difference between precision and recall results in a lower *F*1 score, reflecting a trade-off between these two metrics. The *F*1 score formula is given by the following:(6)$$F1\;score=2\cdot \frac{precision\cdot recall}{precision+recall}$$

All other pixel areas were considered true negatives (TN) and were thus not counted. Precision is also called the positive predictive value (PPV) and recall describes the true positive rate [30,32].

## 3. Results

### 3.1. Difference in Spectral Responses

To differentiate between different components in the fish fillet, it is necessary to see if the spectral responses of these components differ enough to distinguish them. Figure 5 displays the mean spectral reflectance response and standard deviation of various components in the fish fillet. It is evident that the fish muscle (pink) and darker nematodes (blue) can be clearly distinguished in the UV region, the visual spectral range, and part of the NIR region (880–970 nm). The nematodes, lighter in color (olive green), are distinguishable from the fish muscle in the UV region and part of the visual spectra (400–570 nm). It is harder to distinguish between the light nematodes and the fish muscle in the higher end of the visual spectra (590–750 nm) and part of the NIR region, except for 880–970 nm. Blood (green) is distinguishable from darker nematodes in the UV region and the NIR region, but it is harder to distinguish between them in the visual spectral range, except at wavelengths 500–550 nm. Skin remnants (red) are distinguishable from the nematodes in most of the visual spectra and the NIR region however, it is more difficult to distinguish between the darker nematodes and the skin in the UV region, as well as in the lower visual region (400–450 nm). Note the spectra of the nematodes are from nematodes that were on or close to the surface of the fish muscle. These results underscore the added value of extending beyond the visual spectra range for the identification of nematodes or other defects.

### 3.2. Sequence Segmentation Model

Figure 6 illustrates the independent operation and combined layering of the two prediction models along with the ground truth for the same image. Model A, represented by a blue layer, identifies six larger areas as nematodes, with one false positive among five true positives. In contrast, model B, represented by a yellow layer, correctly identifies the false positive as a skin remnant. The final nematode segmentation excludes all pixels identified by model B, as well as any overlapping pixels from both models.

Smaller predicted areas in the image, considered noise, are further analyzed in Section 3.3.

### 3.3. Noise Reduction

The model’s prediction masks include numerous noisy pixels. Figure 7 displays the region sizes in the training data’s masks.

Figure 8 demonstrates the impact of varying size filtering thresholds on the *F*1 score at different IoU thresholds, where the IoU threshold indicates the minimum IoU required for a predicted region to be considered a true positive. With increasing IoU threshold, the *F*1 score lowers, reflecting a stricter criterion for classifying a prediction as a true positive. As the IoU threshold rises, only predictions with a higher degree of overlap with the ground truth are considered correct, leading to fewer true positives and potentially more false negatives. This stricter evaluation results in a lower *F*1 score, indicating a trade-off between precision and recall.

The impact of varying size filtering thresholds further illustrates this relationship: as the size filtering threshold increases, it affects the detection of smaller predicted regions, impacting both the precision and recall and, consequently, the *F*1 score. Peak performance, considered optimal in this case, is achieved at a log scale 5 filtering threshold for the training data, as shown in Figure 8a. The *F*1 score initially improves with higher filtering thresholds but declines after surpassing log scale 5. For the test data, in Figure 8b, a similar trend is observed, with the optimal performance occurring at a log scale 5.7 filter threshold. This confirms the conclusions drawn from Figure 7. Removing smaller regions can enhance the prediction accuracy, but excessively large filtering thresholds may negatively impact performance. This demonstrates the sensitivity of the *F*1 score to the balance between rejecting false positives and capturing true positives, especially in the context of object detection tasks like nematode identification in fish fillets.

### 3.4. Impact of Various Size Filtering Thresholds at a Constant IoU

In this study, the goal is to detect nematodes, allowing the setting of an appropriate IoU threshold without excess. By fixing the IoU threshold at 0.1, the effect of different size filtering thresholds on the *F*1 score can be observed. As demonstrated in Figure 9, an overly large filtering threshold detrimentally affects the model’s performance. It should be highlighted that when several predicted regions overlap with the same ground truth area, only the Intersection over Union (IoU) of one of these regions is assessed. After a ground truth region has been marked as a true positive, any additional overlapping predictions are not evaluated for IoU. In such scenarios, with a fixed IoU threshold, at least one of the overlapping predicted regions meets or exceeds this threshold. However, a limitation of this approach is that it does not provide a comprehensive evaluation of the actual IoU between the predictions and the ground truth. Nonetheless, since the primary goal of this study is to maximize nematode detection, the IoU threshold is set appropriately to avoid excessive stringency. Figure 9 also verifies what we see in Figure 8, that log scales 5 and 5.7 filter thresholds achieve the highest *F*1 score, for training and test sets, respectively.

### 3.5. Recall and Precision over Filtering Thresholds

Figure 10 illustrates how the model’s recall and precision vary with different filtering thresholds. This study considers optimal performance as achieving both high recall and precision simultaneously. The preferred balance may vary based on specific circumstances and the emphasis on higher recall or precision. In this context, prioritizing recall might reduce precision and potentially compromise the quality of the fish, while prioritizing precision might lead to lower recall, impacting food safety. The optimal balance for ensuring the best consumer experience of the final product remains undetermined. In the training data, the optimal performance was achieved at a 0.1 × 10^6^ threshold (Figure 10a), with recall at 83% and precision at 83%. For the test data, the best results were obtained at 0.23 × 10^6^ filtering threshold, achieving 88% precision and 79% recall.

To evaluate the effectiveness of the sequence segmentation model, precision and recall metrics were calculated for model A. The results are presented in Figure 11. This provides insights into the contribution of model B to the performance of the sequence segmentation model. For the training data, the optimal performance was achieved at 0.2 × 10^6^ filtering threshold, with precision at 60% and recall at 74%. For the test data, the optimal performance was achieved at a similar filtering threshold, with precision at 50% and recall at 80%.

## 4. Discussion

Our results indicate that using MSI images with the normalized Canonical Discriminant Analysis (nCDA) can be useful in the next steps of the automation of nematode detection in white fish fillets. The proposed method was able to achieve 88% precision and 79% recall for our manually annotated test data. This performance demonstrates the model’s ability to effectively identify nematodes, fulfilling the key objective of a well-performing model while concurrently controlling the false positive rate to a reasonable level. The significance of these findings lies in striking a delicate balance between minimizing false negatives and false positives. This balance is paramount in fish processing, where both types of errors can have substantial consequences for food safety and processing efficiency. An early study using MSI for nematode detection emphasized that a combination of NIR and visible spectral range would be the best choice for nematode detection [33]. Similar findings have been reported [19,20] demonstrating that such spectral combinations could detect nematodes deeper within the fillet than manual inspection, although both studies showed a relatively high false positive ratio. The current study showed similar results regarding the detection of nematodes of darker color, while the UV range was more appropriate for the differentiation of lighter nematodes from the white fish muscle. This underscores the importance of including wavelengths from the UV, visual, and NIR ranges during nematode detection.

Sivertsen et al. (2012) [20] developed an HSI system capable of detecting nematodes in cod fillets on a moving conveyor belt, and their system demonstrated a notable detection capability. The overall detection rates reported were 61.5% for all nematodes, 60.3% for pale nematodes, and 70.8% for dark nematodes. However, 60% of the fillets had at least one false positive. Since then, the system has been further developed and preliminary testing indicates that the system can be operated at industrial speed for detecting nematodes at a conveyor belt moving at up to 400 mm/s, with a high detection rate and almost no false positives [34]. How high the detection rates and number of false positives were, however, not stated in said preliminary results. False-negative results are, furthermore, more concerning from a food safety point of view, since undetected nematodes can pose considerable health risks [35], highlighting that further developments in the technology are needed.

The comparison of these results with our findings highlights both the challenges and the advancements in nematode detection technologies. While the approach of Sivertsen et al. (2012) [20] achieved substantial detection rates, the high false alarm rate in a significant proportion of samples underscores the difficulty in distinguishing nematodes from other features in the fillets. Conversely, our study, which combined MSI with nCDA, attained higher precision and recall rates. This suggests an enhanced ability to identify nematodes with reduced false positives. However, it is important to recognize that our image acquisition was conducted in a controlled laboratory setting with pre-cut fillets, and not on a moving conveyor belt. While hyperspectral imaging in industrial settings offers faster processing times and advancements in illumination have significantly increased the signal-to-noise ratio, the detection rate remains influenced by storage time and sample handling conditions [20,36]. This difference in conditions is a crucial factor to consider when interpreting the comparative effectiveness of these methods. This analysis not only reflects the progress made in this field but also emphasizes the need for ongoing research and development in imaging and analysis techniques to further improve the efficiency and accuracy of nematode detection in fish processing. This study’s limitations, notably the potential under-detection of nematodes and the small dataset size highlight significant areas for future research and model enhancement. The current approach, constrained by labeling only nematodes that were visibly detected on the MSI images, may not fully capture nematodes embedded deeper within the fillets that could have been found with an exhaustive visual inspection or with the two standardized ISO methods (UV press and artificial digestion) [14,15]. Visual inspection is still the official, non-destructive method used during processing today to locate nematodes [10]. However, research shows varying results regarding the effectiveness of visual inspection, especially regarding the detection rate of *A. simplex sensu stricto*. Furthermore, additional adaptations are needed for detecting nematodes in other fish species. Only 7 to 10% of *A. simplex sensu stricto* larvae in the fillets of herring, mackerel, and blue whiting were detected by candling [37]. This further emphasizes the necessity for more research to establish consistent and accurate non-destructive methods for nematode detection in diverse fish species [16] to provide a reliable benchmark for comparing automatic solutions, including standardization for hyperspectral/multispectral imaging [11]. Another benefit of applying MSI/HSI for nematode detection is the possibility of simultaneously analyzing several other quality and safety parameters of the fish muscle [34]. Quality and safety parameters that MSI/HSI can assess include species recognition (fraud prevention) [38], freshness [39,40], proximate composition [41], physical parameters (such as muscle color, texture, etc.) [42,43], and microplastics detection [44], to mention a few.

Factors such as the depth at which nematodes can be embedded in the fillet certainly still pose a challenge, especially considering the limitations of MSI technology in detecting such deeply embedded parasites. Currently, our method can detect nematodes embedded up to 8 mm into the fish fillet [19]. The detection of *A. simplex sensu stricto* characterized by their small size and almost clear coloration, thus presents another significant challenge [45], especially when deeply embedded within the muscle. In comparison, the official methods—UV press and artificial digestion—have shown high accuracy and sensitivity for detecting *Anisakidae* L3 in fish fillets, with studies indicating 100% and 98% accuracy, and 100% and 96% sensitivity, respectively [16]. However, as previously mentioned, these methods are destructive and do not optimally fit into online processing, highlighting the need for further development of non-destructive methods for the task.

Future work must address these challenges to improve model efficacy. Expanding the dataset is critical. The current dataset, consisting of only 270 images, may limit the model’s ability to generalize effectively images with diverse characteristics, potentially affecting its robustness and applicability in broader contexts [46]. This expansion should include a wider array of images, capturing nematodes at various depths, locations, and sizes to enhance the model’s learning and adaptability. Furthermore, implementing post-imaging validation, where fillets undergo detailed inspections after MSI imaging or other standard ISO methods, could significantly refine detection accuracy for less visible nematodes. To enhance the robustness of our method and to address variations in model performance, we standardized the distance from the fillet during imaging. This approach is necessary because the Intersection over Union (IoU) threshold is calibrated for a constant distance. The importance of IoU in image segmentation and object detection has been well-documented in existing literature [47,48]. As a result, the current method is not inherently scale-invariant. To improve this, adjusting the IoU threshold based on the imaging distance and camera resolution could be a beneficial strategy. This modification aims to make the detection process more adaptable, ensuring it remains effective across different scales and resolutions. By implementing these adjustments, our method could achieve scale and resolution invariance, enhancing its applicability in diverse conditions. Also, adapting the model to accommodate varying camera resolutions could ensure consistent performance across different imaging setups.

Our results, which show high precision and recall in nematode detection, underpin these conclusions by demonstrating that automated detection is not only feasible but effective in enhancing operational efficiency, safety, and quality control in fish processing. This represents a notable advancement in the industry, aligning with the increasing trend towards automation and precision in food processing. Furthermore, it supports manufacturers in creating advanced tools for detecting nematodes and other applications.

## 5. Conclusions

This study demonstrates the potential of using Multispectral Imaging combined with normalized Canonical Discriminant Analysis for the automatic detection of nematodes in Atlantic cod fillets. The proposed sequence segmentation model achieved promising results with 88% precision and 79% recall on the manually annotated test data, highlighting its ability to effectively identify nematodes while maintaining a reasonable false positive rate and false negative rate.

However, the study also acknowledges its limitations, including the potential under-detection of deeply embedded nematodes, the difficulty in identifying small, clear-colored nematodes, and the relatively small dataset size. Future research should focus on expanding the dataset, incorporating post-imaging validation techniques, and adapting the model to accommodate variations in imaging conditions.

In conclusion, the application of automatic nematode detection in fish processing holds significant promise for streamlining workflows, enhancing product quality and safety, and achieving cost savings, contributing to the ongoing advancement of efficient and effective nematode detection methods in the fish industry.

## Figures and Tables

**Figure 1 foods-13-02952-f001:**
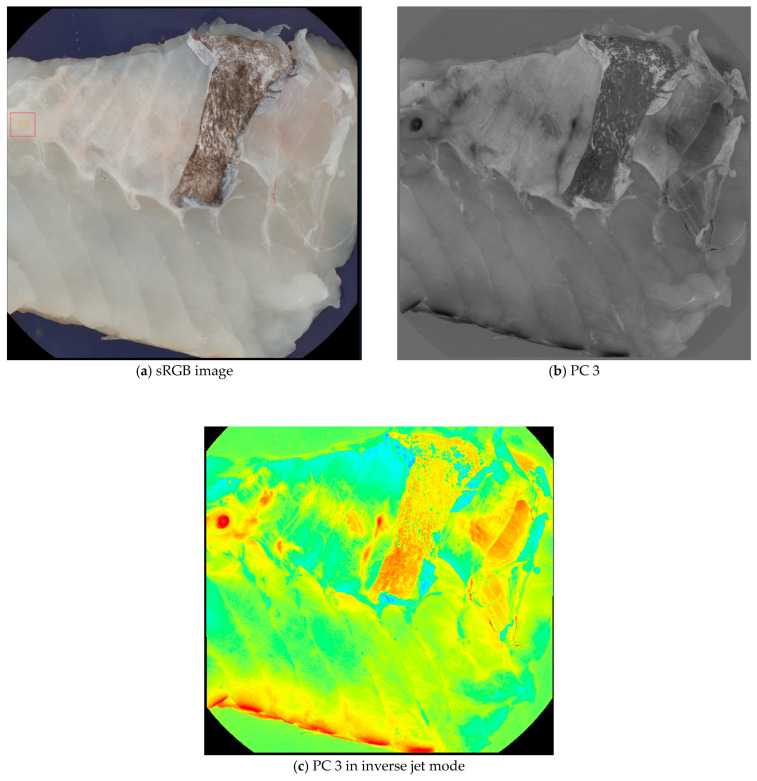
Comparison of a sRGB image (**a**) and a PC 3 (**b**,**c**) images from the data set. Although the nematode in the upper left corner (seen within the red box) is visible in the sRGB image, it is easier to spot in the PC 3 images.

**Figure 2 foods-13-02952-f002:**
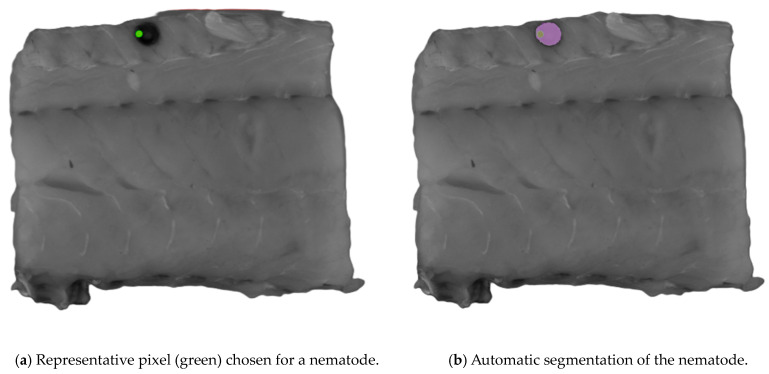
An example of the process of segmenting an object of interest using the Segment Anything Model (SAM). Representative pixels (green) of nematodes were chosen for the automatic segmentation shown in (**a**), where the purple area in (**b**) is the automatic segmented area.

**Figure 3 foods-13-02952-f003:**
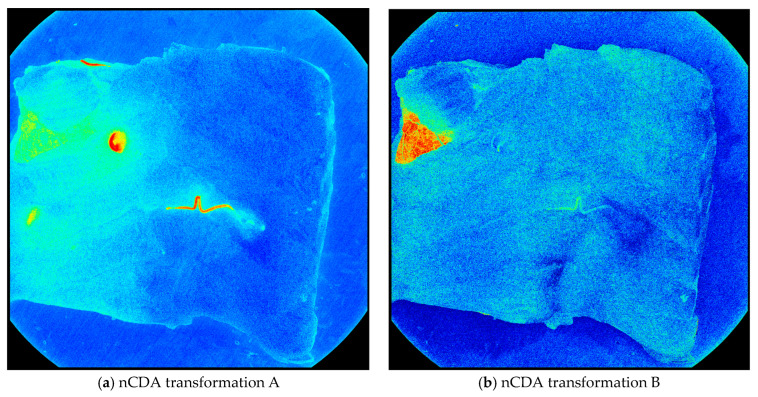
The nCDA transformations applied to an image from the training set. (**a**) Shows nematodes (red) distinguished from other elements (green/blue) and (**b**) shows skin (red) distinguished from every other element (green/blue).

**Figure 4 foods-13-02952-f004:**
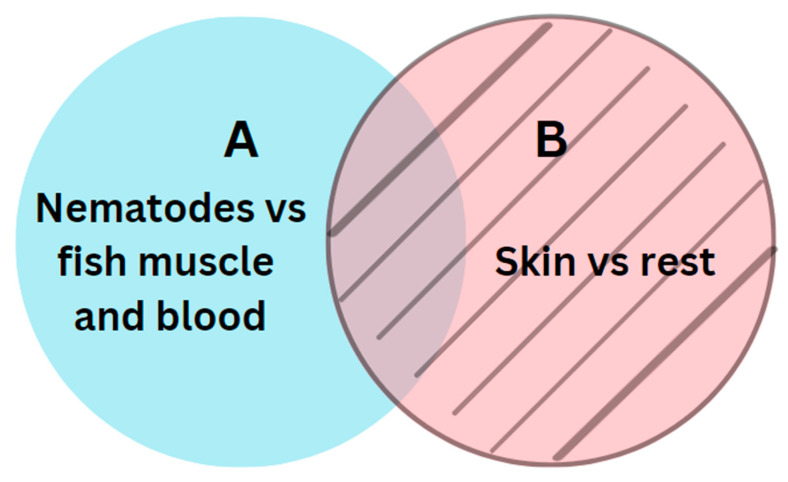
Relations between the two segments A and B.

**Figure 5 foods-13-02952-f005:**
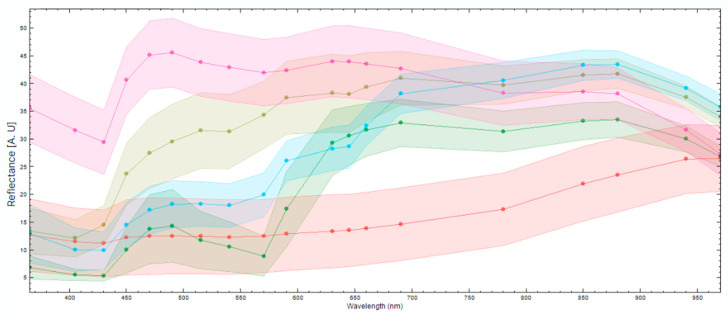
The mean spectral reflectance responses (points) and standard deviation (colored confidence intervals) of different elements found in a fish fillet. The image shows the spectral responses for fish muscle (pink), dark nematodes (blue), light nematodes (olive green), blood (green), and skin remnants (red).

**Figure 6 foods-13-02952-f006:**
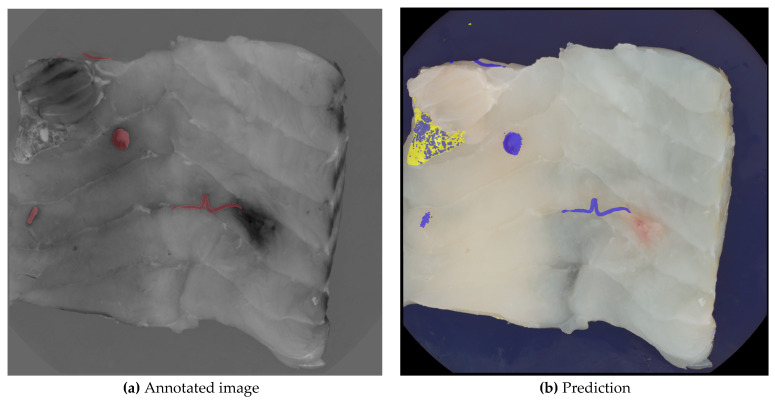
Image (**a**) shows the manually annotated image, where nematodes are marked with red, and image (**b**) shows the layers of the two segmentations. The yellow layer indicates segmentation B, where the skin has been segmented. The blue layer shows segmentation A, where nematodes have been segmented.

**Figure 7 foods-13-02952-f007:**
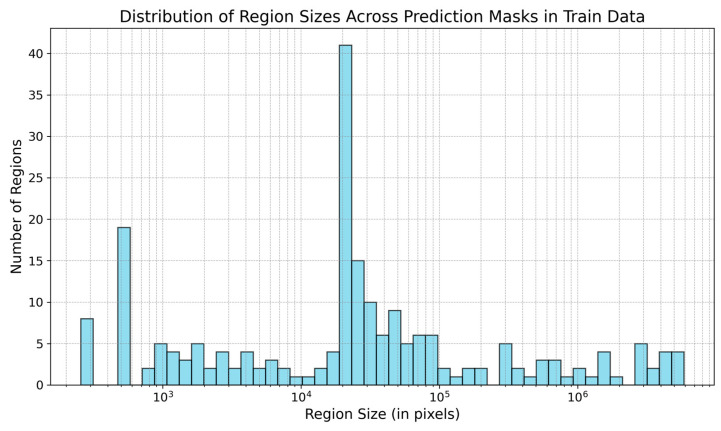
Size of regions (in pixels) present in the prediction masks in the training data. The size of regions is in log scale.

**Figure 8 foods-13-02952-f008:**
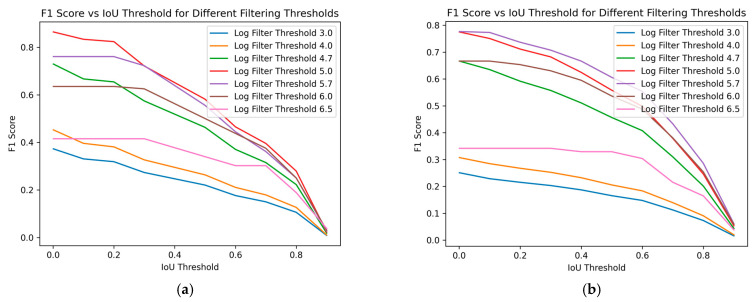
*F*1 score over different Intersection over Union (IoU) thresholds. We can see the performance for different filtering size thresholds (log scale). (**a**) Training data; (**b**) Test data.

**Figure 9 foods-13-02952-f009:**
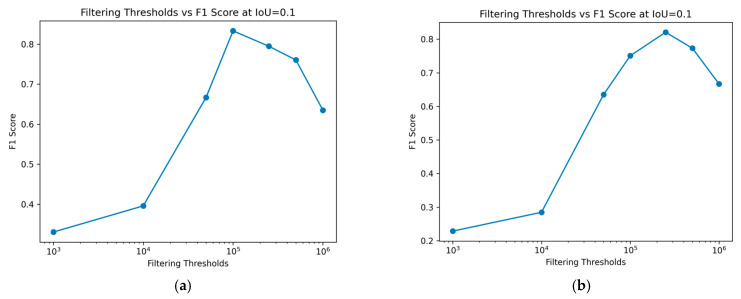
Change in performance, *F*1 score, for different filtering thresholds. (**a**) Training data; (**b**) Test data.

**Figure 10 foods-13-02952-f010:**
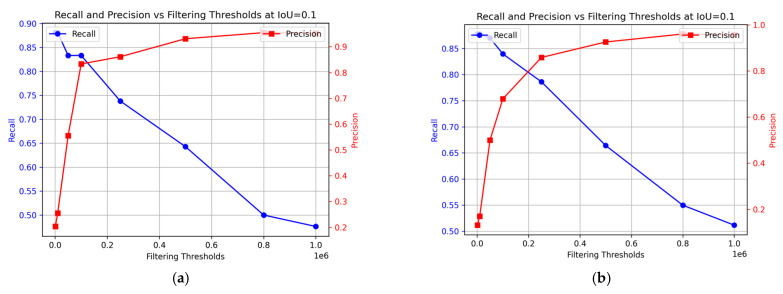
Change in precision and recall at different filtering thresholds for the sequence segmentation model. (**a**) Training data; (**b**) Test data.

**Figure 11 foods-13-02952-f011:**
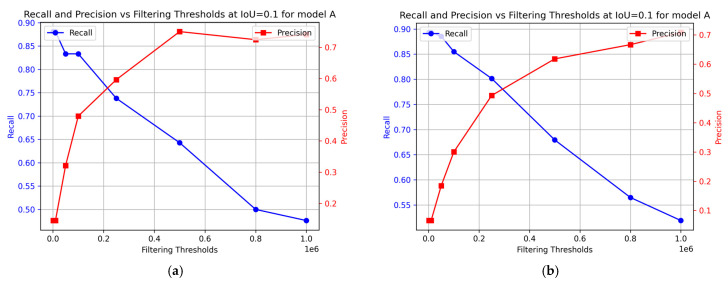
Change in precision and recall as derived from the nCDA analysis at different filtering thresholds for model A. (**a**) Training data; (**b**) Test data.

**Table 1 foods-13-02952-t001:** Distribution of nematodes in parts of fish fillets with the average number of nematodes on each image. The total number of fillets was 50.

Fillet Part	Average No. Nematodes
Loin	0.75
Middle	1.20
Tail	0.35
Belly flap	1.50

## Data Availability

The data presented in this study are available on request from the corresponding author (M.G.).

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
