# Peer review of "Sequence Segmentation of Nematodes in Atlantic Cod with Multispectral Imaging Data"

_foods, 2024, doi:10.3390/foods13182952_

Round 1
Reviewer 1 Report
Comments and Suggestions for Authors
This study used multispectral imaging and Normalized Canonical Discriminant Analysis for segmentation of nematodes in Atlantic cod. The topic fits the scope of this journal. However, some problems should be addressed. Major revisions might be required.
1. Please prepare the manuscript using the MS word templete. Line numbers is missing, which makes it hard to point out which parts should be modified.
2. Since Segment Anything Model was used for segment an object of interest automatically, why was Normalized Canonical Discriminant Analysis necessary?
3. Many deep learning based segemntation methods can be used for semantic segmentation, why were those methods not used?
4. The introduction section was weak. Many outdated references were cited, which were unable to reflect the latest progress at present in this area. What was the challenges in Atlantic cod quality inspection should be summarized.
5. Abstract. 'It minimizes the unnecessary inspection of fillets in good condition and concurrently boosts product safety and quality.' This sentence was hard to understand. Please clarify the purpose of segmentation of nematodes. If nematodes detected, will they be removed during the process? Or just determine if there are nematodes or not?
6. Figure 1, why PC3 was selected?
7. Figure 2. It was hard to understand. What was the definition of the red line? Where was the segmentation results?
8. Figure 5, what was the definition of the y-axis?
9. ‘using 54 images as training data, and 216 images for test
data.’ Why did such a training/testing ratio used? A validation dataset was missing, how to perform hyper parameter optimization or how to stop training?
10. It was a semantic segmentation task, IoU can be the most suitable performance indicator. Please metion it the abstract and conclusion.
11. In the discussion section, quite a lot references before 2015 was used. The novelty of this study cannot be proved by comparing with these outdated cases.
Comments on the Quality of English Language/
Reviewer 2 Report
Comments and Suggestions for Authors
very interesting paper. First of all authors need to reformat the paper according to foods template, we all do.
Secondly, please label all equations used and be careful with the symbols. Please explain some of the strange, not well known symbols, eg. before 2.5 and cite them in the text.
Besides performance metrics I need to see an ANOVA and test for significance.
Some of the images need to be in colour eg. Fig. 1 and 2.
Fig. 10 and 11 are derived from n-CDA analysis? Please mention this.
Discussion needs to be strengthened.
Round 2
Reviewer 1 Report
Comments and Suggestions for Authors
Thank you for your response. All the concerns have been addressed satisfactorily.
Comments on the Quality of English Language/
Reviewer 2 Report
Comments and Suggestions for Authors
authors have replied sufficiently and paper can be accepted